# Fouling-resistant biofilter of an anaerobic electrochemical membrane reactor

Qilin Yu [1] & Yaobin Zhang [1]*

Membrane fouling is a considerable challenge for the stable operation of anaerobic membrane-based bioreactors. Membrane used as a cathode is a common measure to retard fouling growth in anaerobic electrochemical membrane bioreactors (AnEMBR), which; however, cannot avoid the fouling growth. Here we report a strategy using the membrane as an anode to resist membrane fouling in an AnEMBR. Although aggravating in the initial stage, the fouling on the anode membrane is gradually alleviated by the anode oxidation with enriching exoelectrogens to finally achieve a dynamic equilibrium between fouling growth and decomposition to maintain the operation stable. A mesh-like biofilter layer composed of cells with less extracellular polymeric substance (EPS) is formed on the membrane surface to lower the trans-membrane pressure and promote the interception of the anode membrane. The membrane has high electron storage and transfer capacities to accelerate the oxidation of the intercepted fouling materials, especially, the redundant EPSs of the biofilter layer.

[1] Key Laboratory of Industrial Ecology and Environmental Engineering (Dalian University of Technology), Ministry of Education, School of Environmental Science and Technology, Dalian University of Technology, Dalian 116024, China. *email: zhangyb@dlut.edu.cn

ntegration of anaerobic digester with membrane bioreactor technology (AnMBR) is an eco-friendly and promising solution for the treatment of wastewater, which leads to clean energy generation and high effluent quality[1]. However, membrane fouling is a considerable challenge for the stable operation of AnMBR, though the produced biogas may sparge the surface of the membrane to alleviate the fouling[2–4]. Plentiful methods have been attempted to resist the membrane fouling[5–7]. AnEMBR, commonly utilizing the inherent biofouling control of the cathode related to electrostatic repulsion and in situ production of biogas ($H_2$ or $CH_4$) via cathode reduction of $H^+$ or $CO_2$, could significantly slow down the formation of sludge cake layer on membrane surface[8]. However, the fouling can still be accumulated on the cathode membrane and plug membrane pore with operation eventually[9,10]. Therefore, developing a method to effectively resist the fouling of anaerobic membrane reactor is highly desirable to maintain its durable operation.

Until now, membranes were rarely utilized as the anode of AnMBR because the electrostatic adherence in the anode aggravates the fouling to decrease effluent flux. However, the sludge cake layer formed on the membrane potentially improves the effluent quality since the fouling layer can actually serve as a filter to intercept small particles[11], just like the sludge cake layer formed on the sand surface of a sand-filter tank, which can make the effluent increasingly clean with operation before backwashing. Once the fouling is maintained to a certain extent to intercept particles and ensure the sufficient water flux through the cake layer, the pore size of membrane is not required to be quite small to directly intercept the particles, but can be amplified to lessen the fouling that blocks in the pores.

Complex organics such as proteins and polysaccharides have been reported to be decomposed in the anode of the bioelectrochemical system via microbial anodic oxidation[12–14]. During this process, electrons produced from oxidation of organics by exoelectrogens are extracellularly transferred to the anode for electricity generation. The manners of extracellular electron transfer of exoelectrogens include direct contact of extracellular nanowires such as pili, electron shuttle, and so on[15,16]. As is well known, proteins and polysaccharides are the main constituents of the extracellular polymeric substances (EPS), and EPS as well as microbial cells contribute a majority of the membrane fouling of MBR[17,18]. Therefore, using the membrane as the anode is likely to decompose the membrane fouling via the microbial anode oxidation, and once the decomposition rate is approximately equal to or higher than the fouling growth rate, the fouling would not be accumulated and even alleviated to enable the stable operation of the AnMBR. Meanwhile, the cells and residual EPS on the anodic membrane are expected to form a biofilter layer with less resistance to purify the effluent.

In this study, a membrane with large apertures is applied as the anode of an AnMBR to investigate its antifouling performance. We expect that the sludge cake layer formed on the anode membrane surface functions as a fouling-resistant biofilter to intercept particles of the effluent, and the mechanisms of fouling decomposition and fouling roles are explored.

## Results

**Trans-membrane pressure profiles along the operation.** The continuous operation of the membrane reactors increased trans-membrane pressure (TMP) and decreased the permeate flux due to the membrane fouling. The obvious vibration of TMP observed in the three reactors (Fig. 1a) could be attributed to the agitation of biogas produced from digestion and the biodegradation of fouling, which might in part alleviate membrane fouling though their effects were weak and unable to maintain the operation

R-M: Membrane without potential applied
R-0.3: Membrane with −0.30 V potential applied
R-0.2: Membrane with −0.20 V potential applied

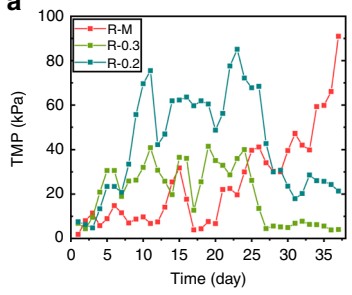

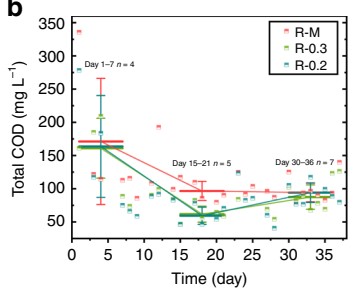

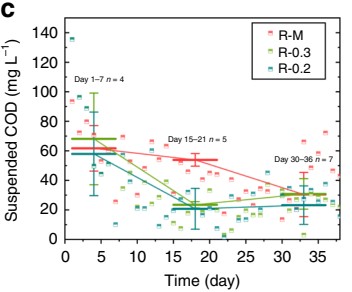

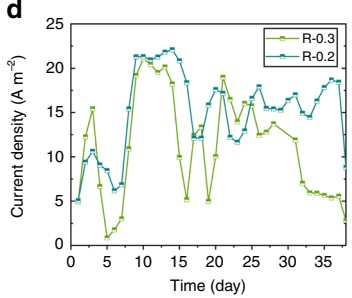

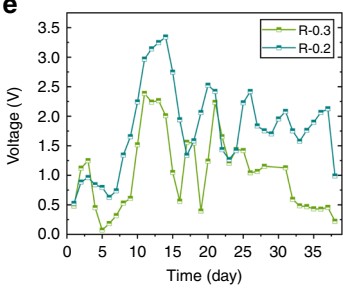

stable. Especially, the electrostatic adsorption and EPS secretion with the electric stimulation further intensified the fouling, leading to faster increases of TMP of the two electro-membrane reactors (R-0.3 applied with −0.3-V anode potential and R-0.2 applied with −0.2-V anode potential) in the initial days. Accordingly, on day 11, the TMP of the two electro-membrane

**Fig. 1** Reactor performances along operation. **a** TMP, **b** total COD, **c** suspended COD, **d** current density, and **e** voltage profiles of the control reactor (R–M: membrane without potential applied) and the electro-membrane reactors R-0.3 (membrane with −0.30-V potential applied) and R-0.2 (membrane with −0.20-V potential applied). (Numbers that follow the ±signs are standard deviation (SD) in this study. Error bars in (**b**) and (**c**) are from Day 1 to 7 (n = 4), from Day 15 to 21 (n = 5), and from Day 30 to 36 (n = 7). Source data are provided as a Source Data file.)

reactors (R-0.3 and R-0.2) was 40.90 and 75.53 kPa, respectively, compared with 6.78 kPa of the control membrane reactor (R–M with no potential applied). Afterward, TMP of the two electro-membrane reactors began to decrease from day 25 and maintained at low levels that averaged at 5.52 ± 1.15 kPa for R-0.3 and 26.49 ± 6.41 kPa for R-0.2 during days 27–37, respectively (numbers that follow the ± sign are standard deviation (SD) in this study). The decrease in TMP suggested that cleansing fouling with anodic potentials was gradually enhanced to exceed the fouling growth. The TMP at low levels indicated the equilibrium between the cleaning and the growth of the fouling on the anode membrane. Contrary to the two electro-membrane reactors, the overall TMP of the R–M presented a significant upward trend from day 20, and eventually exceeded the threshold to cause breakdown finally.

The stable TMP of the electro-membrane reactors was likely due to anodic oxidation of substrates adsorbed on the membrane that outcompeted the fouling accumulation. In agreement, it has been extensively reported that the anode oxidation intensified the decomposition of organic substrates, including complicated substrates like EPS. The reduction of the fouling might be a result of gradual enrichment of exoelectrogens on the membrane, which drove the anodic oxidation of organic matters to remit the fouling and finally maintained the operation stable. The TMP of the R-0.3 was lower than that of the R-0.2 over the operation, indicating that the membrane with the lower anode potential applied was safer for operation.

**Interception performance of the membranes**. The electro-membranes promoted the total chemical oxygen demand (TCOD) removal with a lower COD concentration in the effluent (Fig. 1b). With the increase in the TMP, the TCOD of the R–M decreased from 171.20 ± 94.98 mg L$^{-1}$ (from day 1 to 7, n = 4) to 96.61 ± 14.47 mg L$^{-1}$ (from day 15 to 21, n = 5), which was higher than that of the electro-assisted reactor (R-0.3: from 161.02 ± 45.05 to 61.70 ± 12.04 mg L$^{-1}$ and R-0.2: from 163.66 ± 76.68 to 59.29 ± 12.93 mg L$^{-1}$). After day 30, the three reactors reached the approximately same TCOD at the end of the operation. It indicated that the higher TMP of R–M (Fig.1a) was beneficial for the sludge cake layer to intercept particles, and the sludge cake layer formed on the membrane actually acted as a filter to intercept particles. From Fig. 1c, the average suspended COD of the two electro-membrane reactors (R-0.3 and R-0.2) decreased from 68.11 ± 31.06 and 57.95 ± 28.29 mg L$^{-1}$ (during days 1–7, n = 4) to 23.48 ± 2.13 and 20.77 ± 13.84 mg L$^{-1}$ (during days 15–21, n = 5), respectively, along with TMP increasing during the same periods. Similarly, the suspended COD of the control reactor R–M also decreased with increasing TMP, but at day 38, the suspended COD of the R–M increased sharply to 102.4 mg L$^{-1}$, indicating the breaking up of the membrane. However, the decrease in TMP of the two electro-membranes (Fig. 1a) did not bring down their interception performances, but the membranes maintained high removals of suspended COD. From day 30 to 36, the average suspended COD levels of R-0.3 and R-0.2 were 30.81 ± 10.32 and 23.22 ± 13.03 mg L$^{-1}$ (n = 7),

respectively, compared with 68.11 ± 31.06 and 57.95 ± 28.29 mg L$^{-1}$ in the initial stage of the two reactors (Fig. 1c). The interception of the two electro-membranes was almost synchronous with changes in TMP. Furthermore, a higher TMP or a thicker cake layer was required for R–M to intercept the similar level of suspended COD as the electro-membrane groups. On day 32, the suspended COD of the R–M was 33.54 mg L$^{-1}$ with its TMP at 42.09 kPa, while the similar suspended COD of the R-0.3 (33.11 mg L$^{-1}$) and the R-0.2 (27.09 mg L$^{-1}$) was obtained with the TMP only at 7.75 and 20.26 kPa, respectively. The high interception capacity but lower fouling in the two electro-membrane reactors implied that the composition or structure of the sludge cake layer attached on the membrane surface was likely changed to reserve the filter capacity.

**Electric signal changes of the electro-assisted membrane**. The current density detected between the anode and cathode is a result of the anodic oxidation driven by exoelectrogens[16]. From Fig. 1d, the current density of the two electro-membrane reactors sharply increased from the 6th day, likely due to the enrichment of exoelectrogens to enhance anodic oxidation and electric production. EPS, a main composition of the fouling, could serve as an organic substrate for anodic oxidation. With increasing of the anode oxidation, the fouling was decomposed to thin the sludge cake layer that resulted in the decrease in TMP. The consumption of EPS on the electro-membrane eventually led to the inadequacy of substrates for the anodic oxidation, and correspondingly the electric current density began to decay from day 15. As the cake layer became thinner, the electrostatic adherence of organic matters on the surface of the cake layer on the electro-membrane was enhanced due to the gradient distribution of the electric field. Thus, the adsorbed organics could also serve as the substrate to supplement the inadequate EPS for anode oxidation. Eventually, the fouling growth resulted from the interception, and the fouling cleaning by the anodic oxidation reached the dynamic equilibrium to maintain the operation stable.

As shown in Fig. 1e, the profiles of the voltage of the reactors agreed with the above results. The average voltage between the anode and cathode was about 1.86 ± 0.21 V in the R-0.2, which was higher than that in the R-0.3 (0.48 ± 0.06 V) at the final operation stage (from days 32 to 37, n = 6). It was reported that excessive voltage (higher than 0.80 V) was harmful to the anaerobic microbes[19]. Clearly, the lower voltage of R-0.3 was preferred for the anaerobic microbes.

**Cross-profile and surface morphology of the membranes**. A microscope and scanning electron microscope (SEM) were used to observe the morphologies of the membranes at the end of the operation. The membranes of R–M and R-0.3 were characterized to investigate the effects of anodic potential on the membrane fouling. From the cross section of the membrane in Fig. 2a, b, a thick sludge cake layer (32.20–62.56 μm) was observed on the membrane surface of R–M, while a thin sludge cake layer (18.00–31.50 μm) was obtained with the anode potential applied. The thin sludge cake layer indicated the alleviation of fouling with the potential on the anodic membrane, while the thick sludge cake layer indicated the heavy fouling without the potential. From the SEM analysis (Fig. 2c, d), the cells of the sludge cake layer on the membrane without electro-assistance were encysted with compact EPS-like matters that were huddled together to form dense structures, which might efficiently intercept suspended matters (Fig. 1c) and increase TMP (Fig. 1a). Comparatively, a clear array of sludge cell appeared on the anode membrane, in which the gaps of cells were visible with less extracellular substrates to seemingly form mesh-like sludge cake structures. The

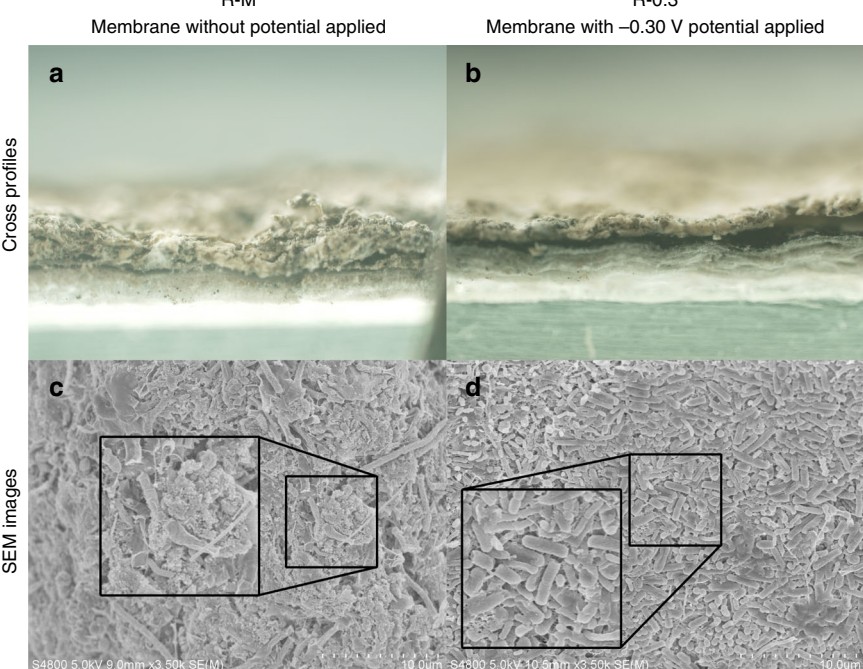

**Fig. 2** Morphologies of the membranes. Cross-profiles of **a** the control membrane (R-M: membrane without potential applied) and **b** the electro-membrane (R-0.3: membrane with −0.30-V potential applied). Surfaces morphologies of the sludge cake layers of **c** the control membrane and **d** electro-membrane. (Membranes were taken out at day 38)

mesh-like structures formed by the well-aligned stacking of cells of the anodic microbes could function as a biofilter membrane to hold back suspended matters to obtain a higher effluent quality as well as maintain a high water flux. It indicated that an anodic potential imposed on the membrane could cleanse the majority of fouling including EPS. Namely, the EPS and suspended matters of the sludge cake layer of the electro-membrane could be biodegraded by the anodic oxidation, which caused the cells with less EPS-like matters or others wrapped to form the mesh-like biofilter to obtain the high filtration efficiency and low TMP.

**Fouling distribution on the sludge cake layers**. Percentages of total cells, proteins, and polysaccharides in the sludge cake layer were detected by confocal laser scanning microscopy (CLSM) (Fig. 3a, b). The fluorescence intensities of total cells (red), proteins (green), β-polysaccharides (blue), and α-polysaccharides (purple) of the electro-membrane (Fig. 3b) were significantly weaker than those of the control membrane (Fig. 3a), showing that the fouling, mainly consisting of cells, proteins, and polysaccharides, was alleviated with the anodic potential. However, from the calculation of the intensities (Table 1), with the anodic potential imposed on the membrane, the intensity percentage of the total cell increased from 25.06 to 38.06%, along with the decrease in the intensity percentages of proteins and polysaccharides surrounding cells, which was well in agreement with the mesh-like cell stacking with less EPS wrapped (Fig. 2). Especially, the proteins and polysaccharides intensity of per cell decreased from 1.17 to 0.56 and 1.82 to 1.07 with the anodic potential, respectively, indicating that EPS such as proteins and polysaccharides was more likely to be decomposed compared with cells. The cells remaining on the sludge cake layer were exposed to form the mesh-like biofilter to improve the water flux and participate in anode oxidation.

**Electrochemical performances of the membranes**. EPS contains the functional groups such as proteins and carbonyl/hydroxyl

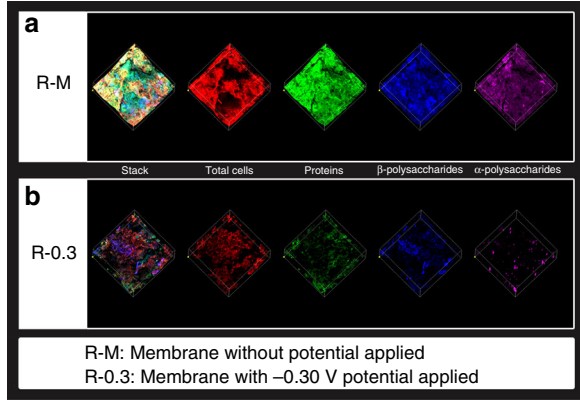

**Fig. 3** CLSM images of **a** the control membrane (R-M: membrane without potential applied) and electro-membrane (R-0.3: membrane with −0.30-V potential applied). (Membranes were taken out at day 38)

that were believed redox-active to store electrons and mediate extracellularly the electron transfer of exoelectrogens[20,21]. Therefore, the decomposition of EPS such as proteins and polysaccharides would decrease its electroactivity for the anode oxidation. Reversely, the sludge cake layer of the electromembrane attained a much higher electron-storage capacity[22] than that of the control membrane from the cyclic voltammogram (CV) curves in phosphate buffer solution (PBS) (Fig. 4a). It seemed that the electroactivity of the sludge cake layer was reserved during the decomposition of sludge cake layer.

From the electrochemical impedance spectrum (EIS) results (Fig. 4b), the resistance of the R–M membrane (including sludge cake layer and carbon nanotube (CNT) membrane) was significantly higher than that of the CNT membrane, while the resistance of anode membrane was approximate with the CNT membrane. The EIS data of three membranes were fitted with a

**Table. 1 Intensities of the total cells and EPSs in the sludge cake layers of the control membrane and electro-membrane**

|  | Total cells | Proteins | β-polysaccharides | α-Polysaccharides | Pn/TC | Ps/TC |
|---|---|---|---|---|---|---|
| R–M | 11.32 (25.06%) | 13.23 (29.30%) | 9.47 (20.96%) | 11.14 (24.67%) | 1.17 | 1.82 |
| R-0.3 | 3.75 (38.06%) | 2.09 (21.20%) | 2.75 (27.93%) | 1.26 (12.80%) | 0.56 | 1.07 |

Control membrane: R–M, membrane without potential applied; electromembrane: R-0.3, membrane with −0.30-V potential applied; membranes were taken out at day 38. Source data are provided as a Source Data file

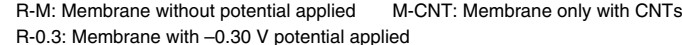

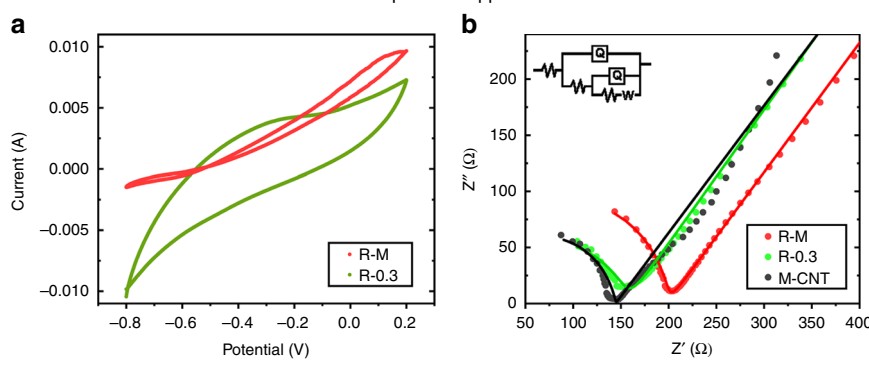

**Fig. 4** Electrochemical performances of the membranes. **a** Cyclic voltammetric analyses of control membrane and electro-membrane in PBS solutions at a scan rate of 5 mV/s. **b** Nyquist plots and the fitted data (solid line) according to the inserted equivalent circuit of CNT membrane (M-CNT: membrane only with CNTs), control membrane (R–M: membrane without potential applied), and electro-membrane (R-0.3: membrane with −0.30-V potential applied). (Membranes were taken out at day 38. Source data are provided as a Source Data file.)

**Table. 2 Resistances of the membranes and conductivities of sludge cake layers of the control membrane and electro-membrane**

|  | Resistance (Ω) | Δresistance (Ω) | Thickness (μm) | Conductivity ($10^{-4}$ S m$^{-1}$) |
|---|---|---|---|---|
| R–M | 198.4 | 54.0 | 32.20–62.56 | 1.86–3.62 |
| R-0.3 | 156.2 | 11.8 | 18.00–31.50 | 4.76–8.33 |

Control membrane: R–M, membrane without potential applied; electro-membrane: R-0.3, membrane with −0.30-V potential applied; membranes were taken out at day 38. Source data are provided as a Source Data file

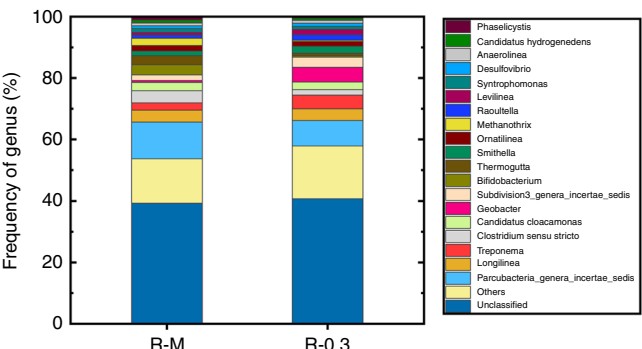

**Fig. 5** Bacterial community structures of the sludge cake layers of the control membrane (R–M: membrane without potential applied) and electromembrane (R-0.3: membrane with −0.30-V potential applied) after operation (at day 38). The genus level with relative abundance lower than 1.00% was classified into group "others". (Source data are provided as a Source Data file.)

simulated equivalent circuit to further depict the electrochemical properties of biofilter cake layer[23]. From the simulated results, the R–M membrane attained a higher membrane resistance (202.5 Ω) than the anode membrane (155.5 Ω) and CNT membrane (146.0 Ω), which meant that the conductivity of the sludge cake layer on R-0.3 was about 1.32–4.48-folds of that on the R–M membrane referring to the layer thickness (Table 2). The higher conductivity of the sludge cake layer on the electro-membrane was beneficial for extracellular electron transfer of anode oxidation[24].

**Effect of anode potential on microbial community**. Geobacter species, a typical exoelectrogen in bio-electrochemical systems[25–27], was specifically enriched in the sludge cake layer of the electro-membrane (Fig. 5). The abundance of Geobacter was 4.80%, about 7.4-folds as high as that of control membrane (0.65%). Geobacter biocatalyzed the decomposition of fouling via the anodic oxidation to maintain the operation stable[16,28]. The enrichment of Geobacter with conductive pili might increase the electroactivity of sludge cake layer such as sludge conductivity[29]. Moreover, the abundance of fermentative bacteria such as Levilinea[30,31] and Raoultella[32] species that could utilize carbohydrates, amino acids, alcohols, lactate,

pyruvate, and fumarate in the electromembrane was also higher than that in the control membrane. The metabolisms of the fermentative bacteria and the cooperation with exoelectrogens forwarded the efficient decomposition of fouling to maintain the electromembrane reactor that performed stably (see Supplementary Table 1).

**Surface group variations and changes in electroactivity**. The sludge cake layers were identified by electrochemical in situ Fourier transform-infrared (FTIR) spectroscopy to reveal the electrochemical activity at the molecular level. Bands that varied with potential shifting were related to redox reactions that occurred at the cell–electrode interface[33]. As shown in Fig. 6a–d, for the sludge cake layer of the control membrane, the intensities

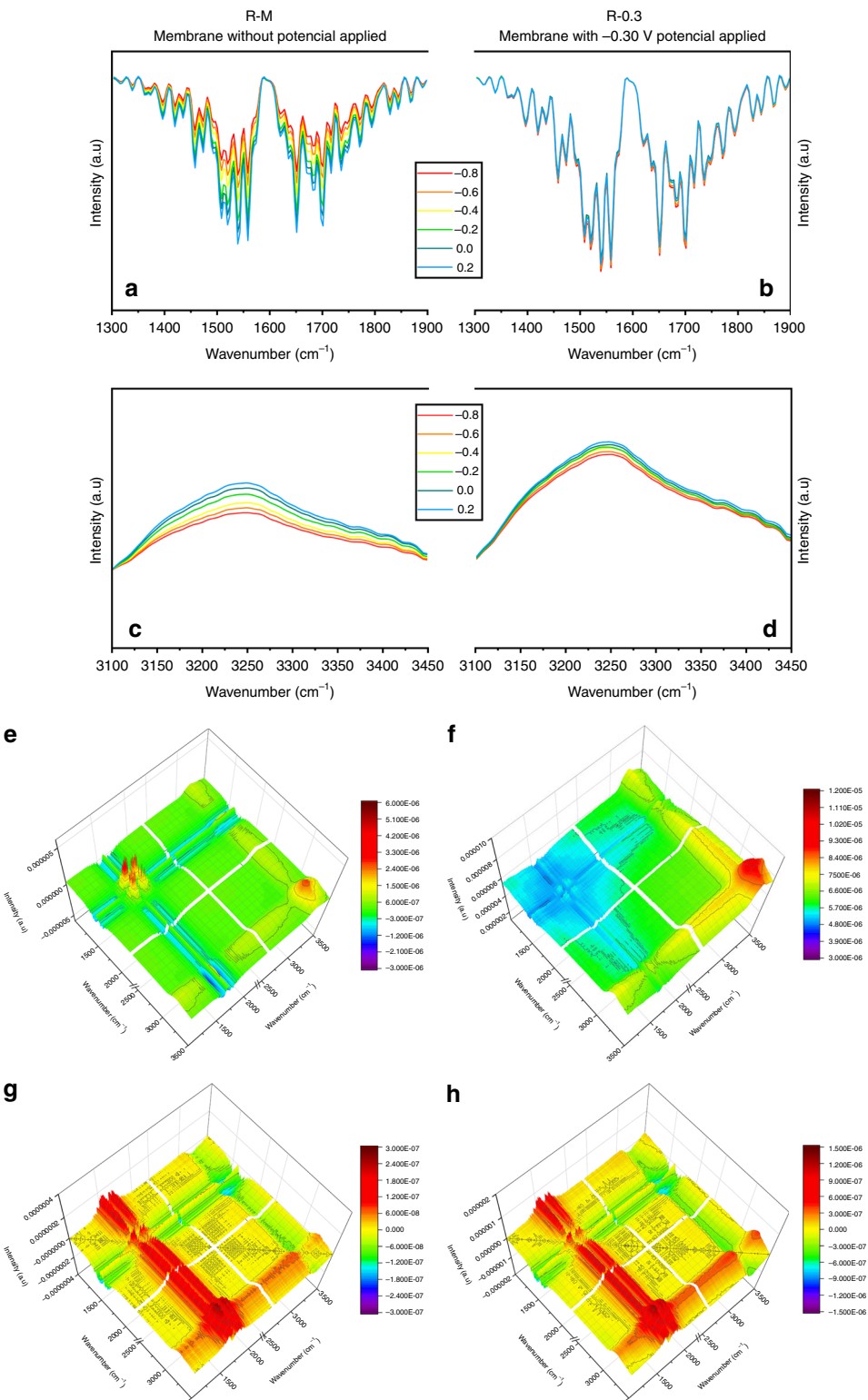

**Fig. 6** Electrochemical in situ Fourier transform-infrared (FTIR) spectroscopy of anode-associated **a** control membrane (R–M: membrane without potential applied) and **b** electromembrane (R-0.3: membrane with −0.30-V potential applied) between 1300 and 1900 cm$^{-1}$, as well as **c** control membrane (R–M) and **d** electromembrane (R-0.3) between 3100 and 3450 cm$^{-1}$ with varying potentials from −0.8 to 0.2 V. Synchronous 2D correlation maps generated from FTIR analysis of anode-associated **e** control membranes and **f** electromembrane. Asynchronous 2D correlation maps generated of anode-associated **g** control membranes (R–M) and **h** electromembrane (R-0.3). (Source data are provided as a Source Data file.)

of the bands around 1550 (stretching vibration of C–N in amide II) and 1650 cm$^{-1}$ (stretching vibration of the C=O group in amide I) increased with the potential shifting from −0.8 to 0.2 V, which was a result of the polarization of C–N and C=O in amide groups, while the decrease in the intensities of the bands around 3300 cm$^{-1}$ (stretching vibration of the N–H group) was a result of the depolarization of N–H[33,34]. The electric susceptibility is proportional to the relative dielectric constant that reflects the electron-storage capacity for a dielectric[35]. Therefore, with the increase in potential from −0.8 to 0.2 V, the polarization intensity of amide groups of sludge cake layer of the control membrane increased, implying that the potential increased the electron-storage capacity of the amide groups of sludge cake layer. Comparatively, the intensities of amide groups of the electro-membrane showed less changes with the potential, but were still higher than the intensities of the control membrane, which suggested that the amide groups of sludge cake layer of the electro-membrane had possessed the residual polarization that was no longer polarized. The remanent polarization of the amide groups in sludge cake layer originated from the anodic potential imposed during the membrane-filter operation, which meant that the sludge cake layer had the higher electron-storage capacity to be involved in the anodic degradation of fouling.

Long-range electron transfer through proteins is a fundamental reaction in energy conversion processes[36–38], and an applied electric field reportedly provided driving force for electron hopping across proteins with redox cofactors[39,40], in which the amide groups and the H bonds can act as relay stations[41]. Therefore, the increased polarization of C–N and C=O band meant that more electrons of these two bands were in high-energy state likely to participate in the electron transfer in which protein with amide groups might act as a momentous mediator. On the other hand, in the proton-coupled electron transfer of proteins (see Supplementary Fig. 1), some protonated side chains of amino acids such as lysine could release H radical from the homolysis of N–H, and then the released H radical was captured by the nearby C=O to form •C–O–H[42–44]. Obviously, the decreased intensity of polarization of the N–H group made the homolysis of N–H easier due to the distribution of electron cloud and static effect, and the increased intensity of polarization of the C=O group also benefited to capture the H radical released from N–H to form the O–H bond. Thus, the decreased intensity of the N–H bond with the potential indicated an enhanced electron transfer within proteins of the sludge cake layer on the electromembrane. Therefore, with the changed bands above, the residual EPS of the electromembrane possessed a higher electroactivity, including the remnant polarization and the electron transfer capacity, although the majority of EPS has been decomposed. From the analysis of 2DCOS, the synchronous maps (Fig. 6e, f) showed that three predominant autopeaks centered around 1550, 1650, and 3300 cm$^{-1}$ for the control membrane, and one predominant autopeak centered around 3300 cm$^{-1}$ for the electro-assisted group, which were consistent with the in situ FTIR spectroscopy. Unobvious peaks presented around 1650 and 1550 cm$^{-1}$ for the electromembrane indicated that the amide groups were no longer sensitive to the potential change. According to the sequential order rules[45], the detailed distributions of the bands (see Supplementary Table 2) and their cross-peaks in asynchronous maps (Fig. 6g, h) indicated that the changes in the amide groups were homologous, which further indicated that proteins with the amide groups were the source of these bands. The changes in the amide groups were nonhomologous with the N–H group, which indicated that the N–H band with the weakened polar intensity was not derived from amide groups, but might be from the side chain of the amide acids to serve as the proton tansfer sites, in agreement with the aforementioned H radical-coupled electron transfer.

In general, the applied potential not only decomposed the EPS, but also improved the electroactivity of the sludge cake layer. Noteworthily, EPS was a mixture of proteins, polysaccharides, nucleic acids, and other components. Proteins such as pili and cytochromes were reportedly capable of participating in extra-cellular electron transfer[46], while some other EPSs such as polysaccharides were rarely reported electroactive. In this study, the enrichment of exoelectrogens with conductive pili, and the improvement of the electron storage and transfer of the proteins could enhance the decomposition of electro-inert EPSs such as polysaccharides on the anode. The alleviation of the fouling further indicated that the redundant EPSs were decomposed in the electromembrane. It also implied that the electro-inert EPSs rather than the modified proteins in the sludge cake layer were the main substrates for anode oxidation. After the operation with anode potential, redundant EPS was decomposed to relieve the membrane fouling, while the electroactive EPS was reserved to maintain an efficient degradation of membrane fouling by anode oxidation. Along with this process, the sludge cake layer transformed to a mesh-like biofilter with efficient anode oxidation capacity (Fig. 7).

**Operation with a complicated wastewater**. A complicated wastewater composed of protein, macromolecular polysaccharide, humic acid, and so on, was fed to further investigate the feasibility of this method. The overall performances of the three reactors in TMP and COD removal were similar to the above-mentioned results with glucose-based wastewater. Specially, after a sharp increase of TMP in the initial days (Fig. 8a), the TMP of R-0.3 and R-0.2 begins to decrease and was finally maintained at low levels (averaging 21.23 ± 2.41 kPa for R-0.3 and 37.20 ± 1.58 kPa for R-0.2, $n = 5$) during days 38–42, respectively. For the R–M, the TMP gradually increased and eventually broke down after day 32. Differently, the complicated wastewater made the fouling growth more rapid, and it took more days (24 days for R-0.3 and 27 days for R-0.2) for the electroreactors to reach the dynamic equilibriums, the TMP levels of which were also higher than the glucose-based feeding. The effluent TCOD and suspended COD (Fig. 8b, c) of the two electroreactors were lower than those of the control, except in the days that R–M suffered from the high fouling that benefited the interception of particles. The electroreactors still maintained the efficient interception capability to improve the effluent quality even with the alleviation of the membrane fouling. The results indicated that the method by using membrane as the anode was also applicable for the complex wastewater and provided a promising strategy for membrane fouling control. Certainly, the operating parameters need to be optimized for the practical use, which warrants further investigation.

## Discussion

In this study, a strategy was proposed to use CNT membrane as the anode to anti-membrane foulings in the operation of AnMBR. The membrane fouling aggravated in the initial stage in the anode membrane reactors due to the electrostatic adherence. However, along with the operation, the decrease in the TMP across the anode membranes indicated the enhanced anode oxidation of the membrane fouling, while the stable TMP states at low levels actually indicated that the fouling growth and decomposition on the anode membranes achieved the dynamic equilibrium eventually. The sludge cake layer with high electroactivity ensured the efficient anode oxidation of the fouling to decompose the excessive EPS. With the massive decomposition of EPS, the sludge cake layer on the anode membrane became thin and promoted the water flux to lower the TMP. The enrichment of exoelectrogens and the induced electroactivity changes of the EPS in the sludge cake layer were essential for the efficient anode oxidation with the

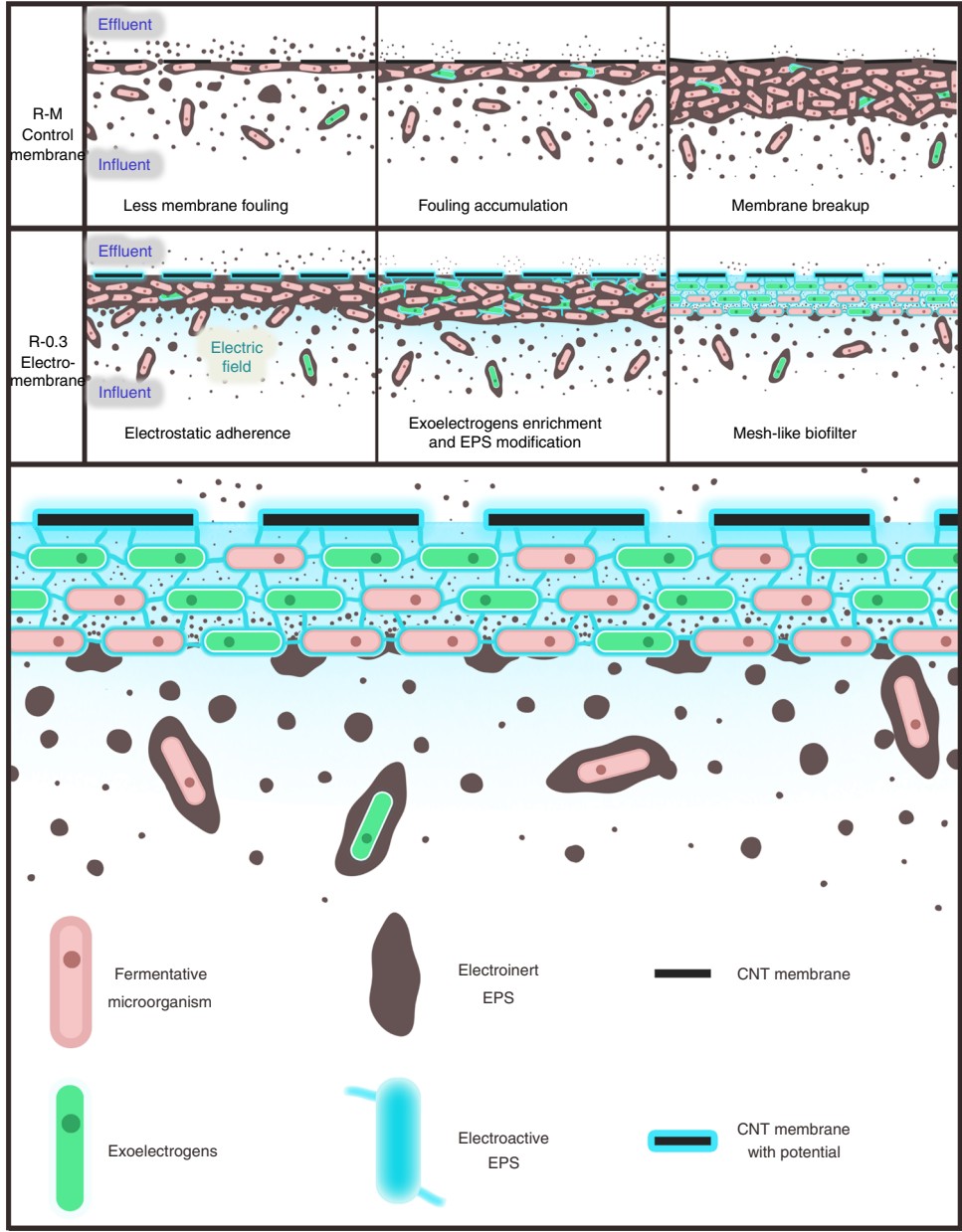

**Fig. 7** Membrane fouling transition profiles and the mesh-like biofilter transformed from fouling in the anode membrane

reduction of EPS, which transformed fouling to a mesh-like dynamic biofilter to possess an excellent interception perfor-mance as well as a high effluent flux (see more details in Sup-plementary Discussion). This study showed the feasibility of using the anode membrane solving the membrane fouling.

## Methods

**AnEMBR setup and operation**. Three anaerobic up-flow anaerobic sludge blanket reactors with 1.0 L working volume ($\Phi 7.0 \times 26$ cm) were applied in this study (see Supplementary Fig. 2). Each reactor was equipped with a CNT membrane module ($1.02 \times 10^{-3}$ m$^2$ working area), six graphite rods ($\Phi$ 1.0 × 6.0 cm), and a silver chloride electrode as the anode, cathode, and reference electrode, respectively. According to the previous studies[47–50], two reactors in which the membranes were imposed with −0.3 and −0.2 V anode potential were referred to as R-0.3 and R-0.2, respectively. The third reactor was not imposed with any potential to be used as the control (referred to as R–M).

The detailed preparation and characterizations of the membrane in terms of pore size distribution and morphology are shown in Supplementary Methods, Supplementary Fig. 3, and Supplementary Fig. 4.

All the reactors were run in parallel over the entire operation (38 days) with a HRT of 16 h and temperature of 38 °C. The seed sludge inoculated in the reactors was collected from an anaerobic digester at a waste sludge treatment plant of Dalian (China). A synthetic glucose-based sewage (see Supplementary Table 3) was fed to the reactors for discussion of the mechanisms, and a synthetic complex sewage (see Supplementary Tables 4–6) was also tested to further verify the feasibility of the antifouling method.

**Operation performance**. COD of the sludge was measured according to Standard Methods for the Examination of Water and Wastewater[51,52]. The current signals were recorded by the electrochemical station (CHI660D, Shanghai Chenhua Ltd., China) that also imposed the anode potential on the membranes. The voltages between the anode and cathode electrode were recorded by a digital gathering module (ZEAL, China). The trans-membrane pressure (TMP) was detected by a paperless recorder (Asmik, China) per second to represent the membrane fouling condition.

**Optical detection**. Confocal laser-scanning microscopy (CLSM) (Fluoview FV-1000, Olympus, Germany) was employed to visualize the distribution of nucleic acids, proteins, and polysaccharides on the sludge cake layers, and the procedures in detail are shown in Supplementary Methods (Sludge Cake Layer Staining). After freeze-drying for 72 h at −50 °C, the thickness of the membranes with sludge cake layer was observed with an optical microscope (Olympus IX83), and the

R-M: Membrane without potential applied
R-0.3: Membrane with −0.30 V potential applied
R-0.2: Membrane with −0.20 V potential applied

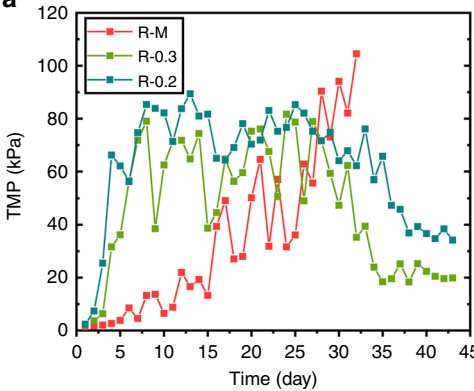

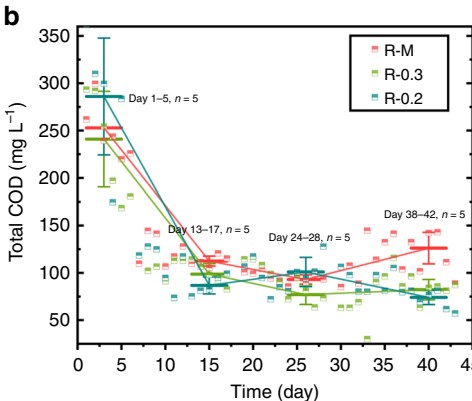

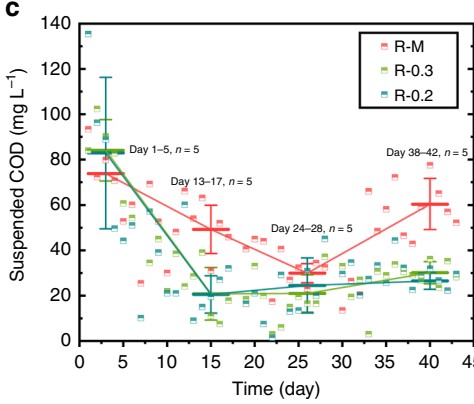

**Fig. 8** Reactor performances along operation. **a** TMP, **b** total COD, and **c** suspended COD profiles of the control reactor (R–M: membrane without potential applied) and the electromembrane reactors R-0.3 (membrane with −0.30-V potential applied) and R-0.2 (membrane with −0.20-V potential applied). (Numbers that follow the ± signs are standard deviation (SD) in this study. Error bars in (**b**) and (**c**) are from Day 1 to 5 ($n = 5$), from Day 13 to 17 ($n = 5$), from Day 24 to 28 ($n = 5$), and from Day 38 to 42 ($n = 5$). Source data are provided as a Source Data file.)

morphologies and structures of the sludge cake layer were also observed by field-emission SEM (FE-SEM, HitachiS-4800).

**Electrochemical measurements**. CV was performed with the workstation (CHI 660, Chenhua Instrument, China) in the range of 0.2 to −0.8 V at scan rates of 5 mV/s in PBS solution, respectively. Electrochemical impedance spectroscopy (EIS) was conducted over a frequency range of 1000 kHz to 0.01 Hz, with a sinusoidal perturbation of 10-mV amplitude, to analyze the internal impedances of the membranes.

**Microbial community**. The procedures and conditions of the microbial community are detailed in Supplementary Methods.

**Outermost membrane surfaces**. The vibrational transition behaviors of the outermost surface of the membrane during electrochemical reactions were characterized by electrochemical in situ FTIR spectroscopy with a FTIR spectrometer (Bruker VERTEX 70). With potentials shifting from −0.8 to 0.2 V as the external perturbation, two-dimensional correlation spectroscopy (2DCOS) was conducted and analyzed. The set and methods in detail are shown in Supplementary Methods (In situ Fourier Transform Infrared Spectroscopy Spectra). The detailed procedures and conditions of in situ FTIR measurement are shown in Supplementary Methods.

**Characterization of chemical and biomolecular materials**. The chemical reagents used in this study were in high purity to ensure the reliability of the results.

**Reporting summary**. Further information on research design is available in the Nature Research Reporting Summary linked to this article.

## Data availability
All data generated or analyzed during this study are included in this published article and its supplementary information files. The source data underlying Figs. 1a–e, 4a, b, 5, 6a, b, 8a–c, Table 1, Table 2, and Supplementary Fig. 3 are provided as a Source Data file. The datasets generated during and/or analyzed during this study are available in the Open Science Framework repository [https://osf.io/u4pwr/].

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

## Acknowledgements

The authors acknowledge the financial support from the National Natural Scientific Foundation of China (21777016 and 51578105) and State Key Research & Development Plan (2018YFC1900901).

## Author contributions

In this study, Yaobin Zhang contributed to the conception and design of the work; Qilin Yu contributed to the acquisition and analysis of data. Yaobin Zhang and Qilin Yu drafted and revised this work together.

## Competing interests

The authors declare no competing interests.
