## [Peer Review File · Nature Communications]

Reviewers' comments:

Reviewer #1 (Remarks to the Author):

In this paper, the authors proposed a new strategy that it uses membrane as anode to resist membrane fouling in an AnEMBR, with the aim of avoiding membrane fouling growth. According to the experimental results, the authors found that a dynamic equilibrium between the fouling growth and decomposition was achieved by this new method. They also revealed that a mesh-like biofilter layer composing of cells and EPS can be formed on the membrane surface, and this biofilter layer have high electron storage and transfer capacities to accelerate the oxidation of the fouling. It seems this new strategy is an effective method for resisting membrane fouling in an AnEMBR. However, there is a fatal flaw that the contents of synthetic wastewater (Table S2) are abnormal in this paper and it is unreasonable and unrepresentative. As shown in Table S2, there are only glucose, NaHCO₃, NH₄Cl, and KH₂PO₄ in the synthetic wastewater, other constituents are also needed according to the published papers. I have grave doubts about the experimental results based on this kind of wastewater, since the formation of membrane fouling usually depends on the characteristics of wastewater. Moreover, the findings of this paper can not be compared with other peer studies based on this kind of wastewater. Therefore, it can not be acceptable for publication in current form in Nature Communications.

Special comments:

- 1) Lines 31-32, "...this biofilter layer had high electron storage and transfer capacities to accelerate the oxidation of the fouling especially EPS...", if a substance has a high electron storage and transfer capacity, how can accelerate the oxidation of this substance? The authors should give more specific explanations.
- 2) Line 121, the section of interception performance of the membranes, the authors should provide some information on the treatment efficiency of the three reactors (R-M, R-0.2, R-0.3), since the COD of synthetic wastewater is high (2000 mg/L).
- 3) Lines 351-355, "...The aggravation of the membrane fouling was observed in the initial stage in the anode membrane reactors. Afterwards, the TMP across the anode membranes went down to low levels, and the fouling growth and decomposition on the anode membranes achieved the dynamic equilibrium eventually..." The authors should modify these sentences, because they are phenomena and should not be present in the Conclusion.
- 4) Lines 370-372, the authors should give some references or reasons for choosing the -0.3 V and -0.2 V anode potential in their experiments.

Reviewer #2 (Remarks to the Author):

This manuscript reports a good piece of research regarding an important novel area such as anaerobic electrochemical reactors. The understanding of fouling when the membrane is used as anode is of great interest to the scientific community. A well work has been made studying fouling, bacteria communities, and the behavior of the membrane. More work could be developed in the future to study the anaerobic process beyond just assessing COD evolution, as well as increasing treatment time and characterizing the hydraulics of the reactor. The paper has been well written and is clearly understood. Only a few comments to make. First, review the text for some misprints (page 20, line 350 "stratagy", for example). In the final publication, be careful with the graphical presentation. In the merged text, as I have received them, they are very difficult to see. The legend of figures and tables must be self-explanatory (for example, in Figure 1 the meanings of RO2 and RO3 are not clear; in Table 1, it could be good to indicate the moment when the sample was taken; etc.).

Response to Editor's and Reviewers' comments on manuscript NCOMMS-19-10944

Title: Fouling-resistant biofilter of an anaerobic electrochemical membrane reactor

Authors: Qilin Yu; Yaobin Zhang

Dear Editor and Reviewers:

We would like to thank the Reviewers for their constructive comments, which have helped us to further improve the quality and clarity of the manuscript.

Below are our detailed responses to the reviewers' comments. The comments from the reviewers are in black, and responses from the authors are in blue. Revisions to the manuscript text are indicated in red. Signs like "Answer" and "In Line:" are highlighted in yellow.

Reviewer #1: Review of Manuscript NCOMMS-19-10944

In this paper, the authors proposed a new strategy that it uses membrane as anode to resist membrane fouling in an AnEMBR, with the aim of avoiding membrane fouling growth. According to the experimental results, the authors found that a dynamic equilibrium between the fouling growth and decomposition was achieved by this new method. They also revealed that a mesh-like biofilter layer composing of cells and EPS can be formed on the membrane surface, and this biofilter layer have high electron storage and transfer capacities to accelerate the oxidation of the fouling. It seems this new strategy is an effective method for resisting membrane fouling in an AnEMBR.

Answer:

We greatly appreciated all the suggestions from the Reviewer and had done our best to revise the manuscript.

However, there is a fatal flaw that the contents of synthetic wastewater (Table S2) are abnormal in this paper and it is unreasonable and unrepresentative. As shown in Table S2, there are only glucose, NaHCO_3 , NH_4Cl , and KH_2PO_4 in the synthetic wastewater, other constituents are also needed according to the published papers. I have grave doubts about the experimental results based on this kind of wastewater, since the formation of membrane fouling usually depends on the characteristics of wastewater. Moreover, the findings of this paper can not be compared with other peer studies based on this kind of wastewater. Therefore, it can not be acceptable for publication in current form in Nature Communications.

Answer:

Thanks for the Reviewer's helpful comments. We agree and have conducted a supplementary experiment with a complicated wastewater, the components of which are as below based on previous studies^{1,2,3,4}: peptone and yeast extract powder (as proteins), sodium polymannuronate (as macromolecular polysaccharide), sodium humate, sodium oleate (as fatty acids), mineral elements and vitamins (detailed in **Table. S3 to S5**). The operating conditions of the supplementary experiment were the same as the early submission. The results showed that the performances of the reactors with feeding this complicated wastewater were similar to the results with the glucose-based wastewater (detailed as follows):

In Line: 385-405.

Operation with a complicated wastewater

A complicated wastewater composed of protein, macromolecular polysaccharide, humic acid, et al., was fed to further investigate the feasibility of this method. The overall performances of the three reactors in TMP and COD removal were similar to the mentioned-above results with glucose-based wastewater. Specially, after a sharp increase of TMP in the initial days (**Fig. 8a**), the TMP of R-0.3 and R-0.2 begin to decrease and finally maintained at low levels (averaging 21.23 ± 2.41 kPa for R-0.3 and 37.20 ± 1.58 kPa for R-0.2, n=5) during day 38 to 42, respectively. For the R-M, the TMP gradually increased and eventually broke down after day 32. Differently, the complicated wastewater made the fouling growth more rapid, and it took more days (24 days for R-0.3 and 27 days for R-0.2) for the electro-reactors to reach the dynamic equilibriums, the TMP levels of which were also higher than the glucose-based feeding. The effluent TCOD and Suspended COD (**Fig. 8b and c**) of the two electro-reactors were lower than those of the control except in the days that R-M suffered from the high fouling that benefited the interception of particles. The electro-reactors still maintained the efficient interception capability to improve the effluent quality even with the alleviation of the membrane fouling. The results indicated that the method using membrane as anode was also applicable for the complex wastewater and provided a new strategy for membrane fouling control. Certainly, the operating parameters need to be optimized for the practical use, which warrants further investigation.

In Line: 443-447 (Method).

The seed sludge inoculated in the reactors was collected from an anaerobic digester at a waste sludge treatment plant of Dalian (China). A synthetic glucose-based sewage (**Table. S2**) was fed to the reactors for mechanisms discussion, and a synthetic complex sewage (**Table. S3 to S5**) were also tested to further verify the feasibility of this anti-fouling method.

On the other hand, the complicated wastewater might disturb the subsequent analysis for mechanism exploration. For example, the addition of humic acid could affect the conductivity and/or capacitance measurement of the sludge cake layers⁵, and the addition of the proteins could cause disturbances on the functional group analysis in FTIR spectra, et al. Moreover, in some previous study on membrane fouling

control^{6,7,8}, acetate or glucose was also used as the sole carbon source in the wastewater. To avoid the ambiguous results, the glucose was used as sole organic matter of the wastewater for mechanism exploration parts in this study.

R-M: Membrane without potential applied
R-0.3: Membrane with -0.30 V potential applied
R-0.2: Membrane with -0.20 V potential applied

Figure 8. Reactor performances along operation. (a) TMP, (b) total COD and (c) suspended COD profiles of the control reactor (R-M: Membrane without potential applied) and the electro-membrane reactors R-0.3 (membrane with -0.30 V potential applied) and R-0.2 (membrane with -0.20 V potential applied). (Numbers those follow the \pm sign are standard deviation (SD) in this study.)

Table. S3. Contents of the complicated wastewater fed to the reactors.

Constituents	Contents
Glucose	1200 mg/L
NaHCO₃	1200 mg/L
NH₄Cl	300 mg/L
Sodium oleate	100 mg/L
Peptone	100 mg/L
KH₂PO₄	60 mg/L
Sodium humate	10 mg/L
Sodium polymannuronate	10 mg/L
Yeast extract powder	10 mg/L
Trace mineral solutions	12.5 ml/L
Trace vitamin solutions	5 ml/L

Table. S4. Contents of the trace mineral solutions.

Constituents	Contents
NTA Trisodium Salt (Free acid)	1.5 g/L
MgSO ₄	3 g/L
MnSO ₄ ·H ₂ O	0.5 g/L
NaCl	1.0 g/L
FeSO ₄ ·7 H ₂ O	0.1 g/L
CaCl ₂ ·2 H ₂ O	0.1 g/L
CoCl ₂ ·6 H ₂ O	0.1 g/L
ZnCl ₂	0.13 g/L
CuSO ₄ ·5 H ₂ O	0.01 g/L
AlK(SO ₄) ₂ ·12 H ₂ O	0.01 g/L
H ₃ BO ₃	0.01 g/L
Na ₂ MoO ₄ ·2 H ₂ O	0.025 g/L
NiCl ₂ ·6 H ₂ O	0.024 g/L
Na ₂ WO ₄ ·2 H ₂ O	0.025 g/L

Table S5. Contents of the trace vitamin solutions.

Constituents	Contents
Biotin	2.0 mg/L
Pantothenic Acid	5.0 mg/L
B-12	0.1 mg/L
p-aminobenzoic acid	5 mg/L
Thioctic Acid (alpha lipoic)	5.0 mg/L
Nicotinic Acid	5.0 mg/L
Thiamin	5.0 mg/L
Riboflavin	5.0 mg/L
Pyridoxine HCl	10 mg/L
Folic Acid	2.0 mg/L

References:

- ¹ Ang, W. S. & Elimelech, M., Fatty acid fouling of reverse osmosis membranes: Implications for wastewater reclamation. *WATER RES* **42** 4393 (2008).
- ² Lin, T., Lu, Z. & Chen, W., Interaction mechanisms of humic acid combined with calcium ions on membrane fouling at different conditions in an ultrafiltration system. *DESALINATION* **357** 26 (2015).
- ³ Yang, Y., Qiao, S., Jin, R., Zhou, J. & Quan, X., Novel Anaerobic Electrochemical Membrane Bioreactor with a CNTs Hollow Fiber Membrane Cathode to Mitigate Membrane Fouling and Enhance Energy Recovery. *ENVIRON SCI TECHNOL* **53** 1014 (2018).
- ⁴ Zhou, Z. *et al.*, Size-dependent microbial diversity of sub-visible particles in a submerged anaerobic membrane bioreactor (SAnMBR): Implications for membrane fouling. *WATER RES* **159** 20 (2019).
- ⁵ Wasiński, K., Walkowiak, M. & Lota, G., Humic acids as pseudocapacitive electrolyte additive for electrochemical double layer capacitors. *J POWER SOURCES* **255** 230 (2014).
- ⁶ Werner, C. M. *et al.*, Graphene-Coated Hollow Fiber Membrane as the Cathode in Anaerobic Electrochemical Membrane Bioreactors - Effect of Configuration and Applied Voltage on Performance and Membrane Fouling. *ENVIRON SCI TECHNOL* **50** 4439 (2016).
- ⁷ Katuri, K. P. *et al.*, A Novel Anaerobic Electrochemical Membrane Bioreactor (AnEMBR) with Conductive Hollow-fiber Membrane for Treatment of Low-Organic Strength Solutions. *ENVIRON SCI TECHNOL* **48** 12833 (2014).
- ⁸ Aslam, A., Khan, S. J. & Shahzad, H. M. A., Impact of sludge recirculation ratios on the performance of anaerobic membrane bioreactor for wastewater treatment. *BIORESOURCE TECHNOL* **288** 121473 (2019).

Special comments:

1) Lines 31-32, "...this biofilter layer had high electron storage and transfer capacities to accelerate the oxidation of the fouling especially EPS...", if a substance has a high electron storage and transfer capacity, how can accelerate the oxidation of this substance? The authors should give more specific explanations.

Answer:

As a main composition of membrane fouling, EPS was a mixture of proteins, polysaccharides, nucleic acids and other components^{9,10}. Some proteins¹¹ such as pili and cytochromes were reported to participate in the extracellular electron transfer from cells to electrodes or minerals, while some other EPSs such as polysaccharides were rarely reported to be electro-active and not to be engaged in the extracellular electron transfer of microbial respiration. In this study, exoelectrogens that possessed conductive pili were enriched in electro-membrane, and the electron storage and transfer of the proteins in the sludge cake layer on electro-membrane was improved, which could enhance the electron transfer and accelerate the decomposition of electro-inert EPSs such as the polysaccharides. The alleviation of the fouling (SEM and CLSM results) further indicated that the redundant EPSs were decomposed in the electro-membrane. It also implied that the electro-inert EPSs rather than the modified proteins in the sludge cake layer were the main substrates for anode oxidation. To make it clear, the following explanations are given in the revised manuscript as follows:

In Line: 30-32.

Multiple lines of evidence confirmed that this biofilter layer had high electron storage and transfer capacities to accelerate the oxidation of the fouling especially **redundant EPSs of the biofilter layer**.

In Line: 367-377.

Noteworthy, EPS was a mixture of proteins, polysaccharides, nucleic acids and other components. Proteins such as pili and cytochromes were reportedly capable of participating in extracellular electron transfer⁴⁶, while some other EPSs such as polysaccharides were rarely reported electro-active. In this study, the enrichment of exoelectrogens with conductive pili, and the improvement of the electron storage and transfer of the proteins could enhance the decomposition of electro-inert EPSs such as polysaccharides on the anode. The alleviation of the fouling further indicated that the redundant EPSs were decomposed in the electro-membrane. It also implied that the electro-inert EPSs rather than the modified proteins in the sludge cake layer were the main substrates for anode oxidation.

In Line: 615-616.

⁴⁶ Lee, D. D., Prindle, A., Liu, J. & Stiel, G. M., SnapShot: Electrochemical Communication in Biofilms. *CELL* **170** 214 (2017).

References:

⁹ Xiao, Y. *et al.*, Extracellular polymeric substances are transient media for microbial extracellular electron transfer. *SCI ADV* **3** e1700623 (2017).

¹⁰ Xiao, Y. & Zhao, F., Electrochemical roles of extracellular polymeric substances in biofilms.

2) Line 121, the section of interception performance of the membranes, the authors should provide some information on the treatment efficiency of the three reactors (R-M, R-0.2, R-0.3), since the COD of synthetic wastewater is high (2000 mg/L).

Answer:

The information on the treatment efficiency of the three reactors is given in the revised manuscript as follows:

In Line 125-134.

The electro-membranes promoted the TCOD removal with a lower COD concentration in the effluent (**Fig. 1b**). With increase of the TMP, the TCOD of the R-M decreased from 171.20 ± 94.98 mg/L (from day 1 to 7, $n=4$) to 96.61 ± 14.47 mg/L (from day 15 to 21, $n=5$), which was higher than that of the electro-assisted reactor (R-0.3: from 161.02 ± 45.05 to 61.70 ± 12.04 mg/L and R-0.2: from 163.66 ± 76.68 to 59.29 ± 12.93 mg/L). After day 30, the three reactors reached the approximately same TCOD at the end of the operation. It indicated that the higher TMP of R-M (**Fig.1a**) was beneficial for the sludge cake layer to intercept particles and the sludge cake layer formed on the membrane actually acted as a filter to intercept particles.

Figure. 1b. Total COD profiles of the control reactor (R-M: Membrane without

potential applied) and the electro-membrane reactors R-0.3 (membrane with -0.30 V potential applied) and R-0.2 (membrane with -0.20 V potential applied). (Numbers those follow the \pm sign are standard deviation (SD) in this study.)

3) Lines 351-355, "...The aggravation of the membrane fouling was observed in the initial stage in the anode membrane reactors. Afterwards, the TMP across the anode membranes went down to low levels, and the fouling growth and decomposition on the anode membranes achieved the dynamic equilibrium eventually..." The authors should modify these sentences, because they are phenomena and should not be present in the Conclusion.

Answer:

The description was modified in the revised manuscript as follows:

In Line: 416-422.

The membrane fouling aggravated in the initial stage in the anode membrane reactors due to the electrostatic adherence. However, along with the operation, the decrease of the TMP across the anode membranes indicated the enhanced anode oxidation of the membrane fouling, while the stable TMP states at low levels actually indicated that the fouling growth and decomposition on the anode membranes achieved the dynamic equilibrium eventually.

4) Lines 370-372, the authors should give some references or reasons for choosing the -0.3 V and -0.2 V anode potential in their experiments.

Answer:

According to previous study, anode-respiring bacteria (ARB) was reported to gain the highest current densities at about -0.30 V vs Ag/AgCl of anode potential and exoelectrogens were generally cultured with a higher anode potential than -0.30 V vs Ag/AgCl. Therefore, we choose -0.20 V and -0.30 V vs Ag/AgCl as the anode potential in this study. The relevant description and references has been given into **Line: 437-440.**

According to the previous studies^{47,48,49,50}, two reactors in which the membranes were imposed with -0.3 V and -0.2 V anode potential were referred to as R-0.3, and R-0.2, respectively.

Line: 617-625.

⁴⁷ Yang, G. *et al.*, Anode potentials regulate Geobacter biofilms: New insights from the composition and spatial structure of extracellular polymeric substances. *WATER RES* **159** 294 (2019).

⁴⁸ Torres, C. I. *et al.*, Selecting Anode-Respiring Bacteria Based on Anode Potential: Phylogenetic, Electrochemical, and Microscopic Characterization. *ENVIRON SCI TECHNOL* **43** 9519 (2009).

⁴⁹ O'Brien, J. P. & Malvankar, N. S., A Simple and Low-Cost Procedure for Growing Geobacter sulfurreducens Cell Cultures and Biofilms in Bioelectrochemical Systems: *Geobacter sulfurreducens* : Anaerobic Cell Cultures and Biofilms in Bioelectrochemical Systems. (2016).

⁵⁰ Bond, D. R. & Lovley, D. R., Electricity production by Geobacter sulfurreducens attached to electrodes. *Applied & Environmental Microbiology* **69** 1548 (2003).

Reviewer #2: Review of Manuscript NCOMMS-19-10944

This manuscript reports a good piece of research regarding an important novel area such as anaerobic electrochemical reactors. The understanding of fouling when the membrane is used as anode is of great interest to the scientific community. A well work has been made studying fouling, bacteria communities, and the behavior of the membrane.

Answer:

We greatly appreciated the Reviewer's nice comments.

More work could be developed in the future to study the anaerobic process beyond just assessing COD evolution, as well as increasing treatment time and characterizing the hydraulics of the reactor. The paper has been well written and is clearly understood.

Answer:

Thanks for Reviewer's helpful suggestions. To further understand the mechanism of the formation and transition of the sludge cake layer in the anode membrane, more investigations need to be conducted in our future work, such as the interspecies information transfer and the synergic effect on sludge cake layer transition of water flow and electric field.

Only a few comments to make. First, review the text for some misprints (page 20, line 350 "stratagy", for example).

Answer:

Thanks. We have revised the entire manuscript, and the misprints have been corrected.

In the final publication, be careful with the graphical presentation. In the merged text, as I have received them, they are very difficult to see.

Answer:

We have tried our best to improve the graph quality in the revised version such as the size of the symbols, legends, scales and frame lines.

The legend of figures and tables must be self-explicative (for example, in Figure 1 the meanings of RO2 and RO3 are not clear; in Table 1, it could be good to indicate the moment when the sample was taken; etc.).

Answer:

The more detailed legends have been given in the figures of the revised manuscript, and the sampling time is also added in the figures and tables.

REVIEWERS' COMMENTS:

Reviewer #2 (Remarks to the Author):

The authors have properly addressed the reviewer's comments. The manuscript may be accepted for publication after the revision of minor misprints (for example, legend in figure 1).

Response to Reviewers' comments on manuscript NCOMMS-19-10944A

Title: Fouling-resistant biofilter of an anaerobic electrochemical membrane reactor

Authors: Qilin Yu; Yaobin Zhang

Dear Reviewers:

We would like to thank the Reviewers for the constructive comments, which have helped us to further improve the quality and clarity of the manuscript.

Below are our detailed responses to the reviewers' comments. The comments from the reviewers are in black, and responses from the authors are in blue. Revisions to the manuscript text are indicated in red. Signs like "Answer" are highlighted in yellow.

Reviewer #2 (Remarks to the Author):

The authors have properly addressed the reviewer's comments. The manuscript may be accepted for publication after the revision of minor misprints (for example, legend in figure 1).

Answer:

We greatly appreciated the Reviewer's nice Reminding. We have checked our manuscript again and some misprints like "withou" and "sign" in the legend in Figure 1 have been corrected.